# Towards Uncovering the Role of Incomplete Penetrance in Maculopathies through Sequencing of 105 Disease-Associated Genes

**DOI:** 10.3390/biom14030367

**Published:** 2024-03-19

**Authors:** Rebekkah J. Hitti-Malin, Daan M. Panneman, Zelia Corradi, Erica G. M. Boonen, Galuh Astuti, Claire-Marie Dhaenens, Heidi Stöhr, Bernhard H. F. Weber, Dror Sharon, Eyal Banin, Marianthi Karali, Sandro Banfi, Tamar Ben-Yosef, Damjan Glavač, G. Jane Farrar, Carmen Ayuso, Petra Liskova, Lubica Dudakova, Marie Vajter, Monika Ołdak, Jacek P. Szaflik, Anna Matynia, Michael B. Gorin, Kati Kämpjärvi, Miriam Bauwens, Elfride De Baere, Carel B. Hoyng, Catherina H. Z. Li, Caroline C. W. Klaver, Chris F. Inglehearn, Kaoru Fujinami, Carlo Rivolta, Rando Allikmets, Jana Zernant, Winston Lee, Osvaldo L. Podhajcer, Ana Fakin, Jana Sajovic, Alaa AlTalbishi, Sandra Valeina, Gita Taurina, Andrea L. Vincent, Lisa Roberts, Raj Ramesar, Giovanna Sartor, Elena Luppi, Susan M. Downes, L. Ingeborgh van den Born, Terri L. McLaren, John N. De Roach, Tina M. Lamey, Jennifer A. Thompson, Fred K. Chen, Anna M. Tracewska, Smaragda Kamakari, Juliana Maria Ferraz Sallum, Hanno J. Bolz, Hülya Kayserili, Susanne Roosing, Frans P. M. Cremers

**Affiliations:** 1Department of Human Genetics, Radboud University Medical Center, 6500 HB Nijmegen, The Netherlands; 2Univ. Lille, Inserm, CHU Lille, U1172-LilNCog-Lille Neuroscience & Cognition, F-59000 Lille, France; 3Institute of Human Genetics, University of Regensburg, 93053 Regensburg, Germany; 4Institute of Clinical Human Genetics, University Hospital Regensburg, 93053 Regensburg, Germany; 5Department of Ophthalmology, Hadassah Medical Center, Faculty of Medicine, The Hebrew University of Jerusalem, Jerusalem 91120, Israel; 6Department of Precision Medicine, University of Campania ‘Luigi Vanvitelli’, 80138 Naples, Italy; 7Eye Clinic, Multidisciplinary Department of Medical, Surgical and Dental Sciences, University of Campania ‘Luigi Vanvitelli’, 80131 Naples, Italy; 8Telethon Institute of Genetics and Medicine (TIGEM), 80078 Pozzuoli, Italy; 9Ruth and Bruce Rappaport Faculty of Medicine, Technion-Israel Institute of Technology, Haifa 31096, Israel; 10Department of Molecular Genetics, Institute of Pathology, Faculty of Medicine, University of Ljubljana, 1000 Ljubljana, Slovenia; 11Center for Human Genetics and Pharmacogenomics, Faculty of Medicine, University of Maribor, 2000 Maribor, Slovenia; 12The School of Genetics and Microbiology, The University of Dublin Trinity College, D02 VF25 Dublin, Ireland; 13Department of Genetics, Health Research Institute-Fundación Jiménez Díaz University Hospital, Universidad Autónoma de Madrid (IIS-FJD, UAM), 28049 Madrid, Spain; 14Center for Biomedical Network Research on Rare Diseases (CIBERER), Instituto de Salud Carlos III, 28029 Madrid, Spain; 15Department of Paediatrics and Inherited Metabolic Disorders, First Faculty of Medicine, Charles University and General University Hospital in Prague, 128 08 Prague, Czech Republic; 16Department of Ophthalmology, First Faculty of Medicine, Charles University and General University Hospital in Prague, 128 08 Prague, Czech Republic; 17Department of Histology and Embryology, Medical University of Warsaw, 02-004 Warsaw, Poland; 18Department of Ophthalmology, Medical University of Warsaw, SPKSO Ophthalmic University Hospital, 03-709 Warsaw, Poland; 19College of Optometry, University of Houston, Houston, TX 77004, USA; 20Jules Stein Eye Institute, Los Angeles, CA 90095, USA; 21Ophthalmology, University of California Los Angeles David Geffen School of Medicine, Los Angeles, CA 90095, USA; 22Blueprint Genetics, 02150 Espoo, Finland; 23Department of Biomolecular Medicine, Ghent University, 9000 Ghent, Belgium; 24Center for Medical Genetics, Ghent University Hospital, 9000 Ghent, Belgium; 25Department of Ophthalmology, Radboud University Medical Center, 6525 GA Nijmegen, The Netherlands; 26Division of Molecular Medicine, Leeds Institute of Medical Research, St. James’s University Hospital, University of Leeds, Leeds LS9 7TF, UK; 27Department of Ophthalmology, The Jikei University School of Medicine, Tokyo 105-8461, Japan; 28Institute of Molecular and Clinical Ophthalmology Basel, 4031 Basel, Switzerland; 29Department of Ophthalmology, Columbia University, New York, NY 10027, USA; 30Department of Pathology & Cell Biology, Columbia University, New York, NY 10027, USA; 31Laboratorio de Terapia Molecular y Celular (Genocan), Fundación Instituto Leloir, CONICET, Buenos Aires 1405, Argentina; 32Eye Hospital, University Medical Centre Ljubljana, 1000 Ljubljana, Slovenia; 33Faculty of Medicine, University of Ljubljana, 1000 Ljubljana, Slovenia; 34St John of Jerusalem Eye Hospital Group, East Jerusalem 91198, Palestine; 35Department of Ophthalmology, Riga Stradins University, LV-1007 Riga, Latvia; 36Children’s Clinical University Hospital, LV-1004 Riga, Latvia; 37Department of Ophthalmology, New Zealand National Eye Centre, Faculty of Medical and Health Sciences, The University of Auckland, Grafton, Auckland 1023, New Zealand; 38Eye Department, Greenlane Clinical Centre, Auckland District Health Board, Auckland 1142, New Zealand; 39University of Cape Town/MRC Precision and Genomic Medicine Research Unit, Division of Human Genetics, Department of Pathology, Institute of Infectious Disease and Molecular Medicine (IDM), Faculty of Health Sciences, University of Cape Town, Cape Town 7925, South Africa; 40Department of Pharmacy and Biotechnology, University of Bologna, 40127 Bologna, Italy; 41Department of Medical and Surgical Sciences, University of Bologna, 40127 Bologna, Italy; 42Unit of Medical Genetics, IRCCS Azienda Ospedaliero-Universitaria di Bologna, 40138 Bologna, Italy; 43Nuffield Laboratory of Ophthalmology, Nuffield Department of Clinical Neurosciences, Oxford University, Oxford OX3 9DU, UK; 44Oxford Eye Hospital, Oxford University NHS Foundation Trust, Oxford OX3 9DU, UK; 45The Rotterdam Eye Hospital, 3011 BH Rotterdam, The Netherlands; 46Australian Inherited Retinal Disease Registry and DNA Bank, Department of Medical Technology and Physics, Sir Charles Gairdner Hospital, Nedlands, WA 6009, Australia; 47Centre for Ophthalmology and Visual Science, The University of Western Australia, Nedlands, WA 6009, Australia; 48Datana Solutions, 54-530 Wroclaw, Poland; 49Ophthalmic Genetics Unit, OMMA Ophthalmological Institute of Athens, 115 25 Athens, Greece; 50Department of Ophthalmology and Visual Sciences, Universidade Federal de São Paulo, São Paulo 04023-062, SP, Brazil; 51Instituto de Genética Ocular, São Paulo 04552-050, SP, Brazil; 52Institute of Human Genetics, University Hospital of Cologne, 50937 Cologne, Germany; 53Department of Medical Genetics, Koc University School of Medicine (KUSOM), 34450 Istanbul, Turkey

**Keywords:** maculopathies, macula, retinal, inherited, sequencing, penetrance

## Abstract

Inherited macular dystrophies (iMDs) are a group of genetic disorders, which affect the central region of the retina. To investigate the genetic basis of iMDs, we used single-molecule Molecular Inversion Probes to sequence 105 maculopathy-associated genes in 1352 patients diagnosed with iMDs. Within this cohort, 39.8% of patients were considered genetically explained by 460 different variants in 49 distinct genes of which 73 were novel variants, with some affecting splicing. The top five most frequent causative genes were *ABCA4* (37.2%), *PRPH2* (6.7%), *CDHR1* (6.1%), *PROM1* (4.3%) and *RP1L1* (3.1%). Interestingly, variants with incomplete penetrance were revealed in almost one-third of patients considered solved (28.1%), and therefore, a proportion of patients may not be explained solely by the variants reported. This includes eight previously reported variants with incomplete penetrance in addition to *CDHR1*:c.783G>A and *CNGB3*:c.1208G>A. Notably, segregation analysis was not routinely performed for variant phasing—a limitation, which may also impact the overall diagnostic yield. The relatively high proportion of probands without any putative causal variant (60.2%) highlights the need to explore variants with incomplete penetrance, the potential modifiers of disease and the genetic overlap between iMDs and age-related macular degeneration. Our results provide valuable insights into the genetic landscape of iMDs and warrant future exploration to determine the involvement of other maculopathy genes.

## 1. Introduction

Macular dystrophies (MDs) are a subgroup of inherited retinal diseases (IRDs), which affect the central part of the retina, known as the macula, and the adjacent retinal pigment epithelium (RPE). Ultimately, progressive degeneration of photoreceptors results in central visual impairment and often blindness. Inherited MDs (iMDs) encompass Best vitelliform macular dystrophy [1], North Carolina macular dystrophy (NCMD) [2,3], cone dystrophy (CD), cone-rod dystrophy (CRD) [4] and Stargardt disease (STGD) [5,6], and they are relatively rare; yet, the true prevalence of iMDs is difficult to determine due to extensive clinical and genetic heterogeneity [7,8] and variability in the age of onset. iMDs often develop in childhood or early adulthood; however, those with a late age of onset can manifest similar clinical features to age-related macular degeneration (AMD), a multifactorial MD typically presenting later in life, which can make a definitive clinical diagnosis challenging [9]. It is prudent to identify genetic variants associated with iMDs to differentiate between late-onset iMDs, AMDs and other MD phenocopies, in addition to providing a genetic diagnosis for patients. Moreover, knowledge of genetic variants causing MDs can provide etiological insights into the molecular pathways leading to disease and offer potential therapeutic options for patients.

Previously, incomplete penetrance has been described in several IRD-associated genes. A genotype is fully penetrant when all individuals harbouring that genotype exhibit clinical symptoms by a certain age. A variant exhibits reduced or incomplete penetrance when the allele frequency (AF) is too frequent in the general population versus the patient population and results in individuals who do not manifest clinical signs. Incomplete penetrance can be observed in both dominant and recessive disorders; however, in some instances, it can obscure an autosomal dominant inheritance pattern—for example, when the offspring of family members are affected and the parents themselves are not clinically affected due to reduced penetrance of the disease. The variants c.5603A>T; p.(Asn1868Ile) [10], c.5882G>A; p.(Gly1961Glu) [11] and c.4253+43G>A; p.[=,Ile1377Hisfs*3] [12] in *ABCA4* have high allele frequencies in the general population and become penetrant when in *trans* with a severe *ABCA4* allele, with some exceptions. Although c.2588G>C; p.[Gly863Ala,Gly863del] in *ABCA4* alone is not considered pathogenic [13], the complex allele c.[2588G>C;5603A>T] acts as a fully penetrant allele with a moderate effect with variable phenotypes [13]. *PRPH2* variants c.424C>T; p.(Arg142Trp) [14], c.514C>T; p.(Arg172Trp) [15] and c.623G>A; p.(Gly208Asp) [16] are also suggested to modify disease penetrance, as well as *RP1L1* c.133C>T; p.(Arg45Trp) [17,18,19,20]. Moreover, the *NMNAT1* c.769G>A; p.(Glu257Lys) variant displays non-penetrance when present in a homozygous state [21].

We previously established a high-throughput, cost-effective sequencing strategy to sequence genes and loci associated with MDs and demonstrated that this method is effective in variant identification for MD patients [22]. This method used single-molecule molecular inversion probes (smMIPs) to create a tailored sequencing panel targeting 105 iMD- and AMD-associated genes and non-coding or regulatory loci, known deep-intronic variants and pseudoexons, AMD-associated risk factors and the mitochondrial genome. A smMIPs sequencing approach enables multiple targets of interest to be incorporated into one assay, which can encompass a combination of coding and non-coding regions. Furthermore, this method can be multiplexed and scaled to test hundreds of patients in one sequencing run in a high-throughput manner at a low cost per sample to better understand genetic variants associated with MDs. This approach can aid in the identification of rare genetic variants, allowing for more personalised and effective treatments. Herewith, we further utilise the previously published MD-smMIPs panel [22] to screen an additional 1352 iMD probands and shed light on the missing heritability of iMDs. We report the likely causative genetic variants for 39.8% of patients and highlight at least ten variants with potential incomplete penetrance in iMDs, two of which have not been reported previously.

## 2. Materials and Methods

### 2.1. smMIP Design

A previously published smMIPs panel [22] was used for this study. In brief, the MD-smMIPs panel comprised 17,394 smMIPs to capture the 5′UTR and protein-coding regions of 105 iMD- and AMD-associated genes and non-coding or regulatory loci, known deep-intronic variants (DIVs) and pseudoexons, AMD-associated risk variants and the mitochondrial genome. Each smMIP comprises a 225-nucleotide (nt) capture region flanked by an extension and ligation probe, custom dual-index adapter sequences, a unique randomer to mark sequence reads from a common progenitor molecule and two index primer sequences (barcodes) to enable individual patient barcoding to generate uniquely tagged libraries.

### 2.2. Patient Cohorts

The patients recruited for this study were selected by collaborators from thirty-one international and two national institutes, where DNA from each patient was isolated in each respective laboratory. Informed written consent was obtained at each individual participating institution. iMD and cone-dominated patients were selected if a clinical diagnosis of STGD, a “STGD-like” phenotype, MD or a cone-led retinal degeneration (e.g., CD or CRD) was made by the referring senior ophthalmologist at each institute, based on the patient’s medical history, their family history and a detailed ophthalmological exam. According to the clinical practice and available equipment of each institute, where possible, most of the following tests were evaluated and performed: best corrected visual acuity, colour vision test, visual field test, slit lamp fundus examination, colour fundus imaging, fundus autofluorescence (FAF) imaging, optical coherence tomography (OCT) imaging, microperimetry and/or electroretinography (ERG). A portion of patients had previously undergone pre-screening methods, including an alternative smMIPs sequencing approach, a targeted gene analysis, whole-exome sequencing (WES) or whole-genome sequencing (WGS). Where possible, additional clinical information on probands was collated and used to assess and support genotype–phenotype correlations. Thirteen individuals with a known mitochondrial DNA (mtDNA) variant, either homoplasmic or heteroplasmic at known levels of heteroplasmy, associated with MD as well as other mitochondrial disorders, were designated as positive controls to ensure that the levels of heteroplasmy could be accurately determined from the mtDNA smMIPs data. 

### 2.3. DNA Sample and Library Preparation

Genomic DNA samples were prepared, as described previously [22]. In brief, DNA was diluted to 15–25 nanograms per microliter (ng/μL), and 100 nanograms of DNA was used for library preparation. Samples were prepared in 96-well capture plates, and those with high-molecular-weight DNA underwent a pre-shearing step, incubating the DNA at 92 °C for 5 min to denature DNA. DNA samples with low molecular weight were added to the capture plate, and library preparation followed using the High Input DNA Capture Kit, Chemistry 2.3.0H, produced by Molecular Loop Biosciences, Inc. (Woburn, MA, USA) using protocol version 2.4.1H. An incubation time of 18 h was used for probe hybridisation. Details of library preparation steps were published previously [22]. Overall, 384 libraries (four capture plates) were pooled per sequencing run, incorporating 360 genetically unexplained iMD probands, 20 positive controls and 4 no template controls. The final library pools were denatured according to Illumina’s NovaSeq 6000 System Denature and Dilute Libraries Guide, resulting in final libraries at 300 picomoles.

### 2.4. NovaSeq 6000 Sequencing and Generation of Sequencing Files 

Paired-end sequencing was performed on a NovaSeq 6000 (Illumina, San Diego, CA, USA) platform using SP reagent kits v1.5 (300 cycles) and custom sequencing primers. Raw FASTQ files were processed through an in-house bioinformatics pipeline, as previously described [23]. Random identifiers were trimmed from sequencing reads and placed with the read identifier for later use. Unique read pairs were assigned to samples based on an exact match with the dual 10 nt index primer sequences per patient and were written to a single binary alignment map (BAM) file, while duplicated reads were removed. The number of forward reads was added to the number of reverse reads, and this value was divided by two to generate the number of mapped reads and calculate the overall average smMIPs coverages. Variants were called using UnifiedGenotyper and HaplotypeCaller from GATK (v3.4–46) for each individual sample, followed by merging of the two variant call sets using the GATK combine variants and joint genotyping functions.

### 2.5. Variant Selection

Copy number variants (CNVs) were first analysed, as previously described [22], to detect large deletions or insertions. In the second step, single-nucleotide variants (SNV) and indels in *ABCA4* were prioritised to highlight frequent and/or causative variants that were previously published and listed in the Leiden Open (source) Variation Database (LOVD; https://databases.lovd.nl/shared/genes/ABCA4; accessed on 25 January 2023). Subsequently, known pathogenic DIVs targeted by the MD-smMIPs panel were extracted. All nuclear SNVs and indels were filtered, as previously described [22]. All homozygous and compound heterozygous SNVs and indels with a minor AF ≤0.5%, and all heterozygous variants in genes associated with autosomal dominant retinal dystrophies with a minor AF of ≤0.1%, were assessed in an in-house cohort containing 24,488 individuals with numerous phenotypes and the Genome Aggregation Databases (gnomAD v2.1.1) for control exome and genome populations. Novel splice-altering variants were identified using SpliceAI [24] in the first instance, whereby variants with a predicted delta score (DS) of ≥0.2 (range: 0–1) for at least one of the four predictions (acceptor loss (AL), donor loss (DL), acceptor gain (AG) and donor gain (DG)) warranted consideration. Thereafter, variants were evaluated using Alamut Visual Plus, version 1.7.1 (SOPHiA Genetics) to predict the impact on splicing using in silico tools, as previously described [22]. The Franklin Genoox platform (https://franklin.genoox.com/) was used to determine the American College of Medical Genetics and Genomics (ACMG) classifications of all variants (accessed on 25 January 2023). Those variants classified as class 3, 4 or 5 by the ACMG classification system [25] (i.e., variant of uncertain significance (VUS), likely pathogenic and pathogenic, respectively) were prioritised. Variant calling was performed for mtDNA targets, and variants located in the mitochondrial genome, which were previously reported to be implicated in retinal disease (i.e., m.3243A>G), were manually extracted from variant call format (VCF) files to determine heteroplasmy levels. Segregation analysis was only performed for a small proportion of probands by individual collaborators, although it is an essential criterion according to the full ACMG guidelines; when at least two different rare variants in a single gene were detected in an individual, they were considered present in a bi-allelic state and thus presumed to be compound heterozygous. A final verdict of “very likely solved”, “possibly solved” or “unsolved” was assigned, as previously described [22], considering the allele frequencies, suggested ACMG classification, pathogenicity of the variants, the gene(s) involved and previous reports in online databases.

### 2.6. Penetrance Calculation

We estimated the penetrance for a subset of variants to predict whether a variant found in the iMD cohort in general had reduced penetrance. Aggregated allele frequencies were obtained from the Genome Aggregation Database (gnomAD-ALL AF) genome and exome databases v2.1.1 (https://gnomad.broadinstitute.org/; accessed on 25 January 2023). The penetrance (*P*) of a given variant was calculated using the following formula:$$P\;\left(D|X\right)=\frac{P\left(X|D\right)\times P(D)}{P(X)}$$
where $P\;\left(X|D\right)$ is the frequency of genotype *X* in iMD probands; P(D) is the prevalence of iMDs in the population (i.e., using the estimated prevalence of 1 in 5000); and P(X) is the frequency of genotype *X* in the general population under a Hardy–Weinberg assumption (using gnomAD-ALL AFs).

### 2.7. ABCA4 Splice Assays

The effects of three novel *ABCA4* variants predicted to impact splicing were assessed in vitro by means of splice assays. Two previously described midigene wild-type (WT) constructs BA7 (here used to test c.1451G>A) and BA16 (here used to test c.3329-124G>T) containing *ABCA4* genomic sequences were used [26]. In order to model the effect of c.6817-679C>G, a new minigene BA39 (hg19: intron 49, g.94,458,910–94,462,144) was generated. 

First, the amplified DNA was inserted into a donor vector (pDONR201; Invitrogen, USA) to create entry clones using the primers listed in Appendix A. Sanger sequencing was performed to select one WT clone and one mutant clone. PCR products were validated by agarose gel visualisation, followed by band excision and purification. Site-directed mutagenesis was performed for c.1451G>A and c.3329-124G>T to insert a point mutation into the WT entry vectors. The entry clones were transferred into the Gateway-adapted destination vector pCI-*NEO*-*RHO* and used to transfect human embryonic kidney T (HEK293T) cells, as previously described [27]. Transfections were performed in duplicate using 600 ng plasmid and FuGENE HD reagent (Promega, Madison, WI, USA), as specified in the manufacturer’s protocol. RNA was isolated after 48 h, and cDNA was synthesised from 1000 ng RNA using the iScript cDNA Synthesis kit (Bio-Rad, Hercules, CA, USA).

Reverse transcription (RT)-PCR was performed as follows: 94 °C for 2 min, followed by 35 cycles of 30 s at 94 °C, 30 s at 58 °C and 1 min at 72 °C, with a final extension step of 5 min at 72 °C. For RT-PCR amplification of c.3329-124G>T and c.6817-679C>G, primers in exons 3 and 5 of *Rhodopsin* (*RHO*), present in the midigene system, were used. For RT-PCR of the c.1451A>G variant, a primer in *ABCA4* exon 9 was used with the *RHO* exon 5 primer. The primers used for mutagenesis and RT-PCR are listed in Appendix A. *Actin beta* (*ACTB*) and exon 5 of *RHO* were amplified as expression and transfection controls, respectively. Agarose gel quantification was performed, and the exact sequence of observed RT-PCR products was confirmed by Sanger sequencing of excised gel bands. Following gel electrophoresis, densitometry analysis using Fiji software version 1.53t was performed to quantify the ratios between different RNA products. Only products with more than 15% of total RNA were reported. The Human Genome Variation Society (HGVS) rules were applied to the cDNA, RNA and protein notation of all variants using the MANE transcript NM_000350.3. 

### 2.8. ROM1 Variant Phasing

Given the ultra-rare combination of *ROM1* variants identified in one proband, long-read amplicon sequencing using the PacBio Sequel system (PacBio, Menlo Park, CA, USA) was performed to determine haplotype phasing. A 1.2 Kb region spanning the two variants was amplified by PCR (primers listed in Appendix A), and long-read sequencing was performed using the SMRT sequencing technology, as previously described [28]. HiFi reads spanning the two variants of interest were visualised using Integrative Genomics Viewer (IGV) version 2.4 to determine whether variants were in *cis* or *trans*.

### 2.9. Identifying Levels of Heteroplasmy in mtDNA

Minimal mitochondrial analysis was performed, where only the known m.3243A>G variant implicated in macular pattern dystrophy [29] was extracted from VCF files. As verification to ensure that the levels of heteroplasmy in mtDNA could be determined, 13 control DNA samples from patients with known mtDNA-associated disease, which had previously undergone diagnostic analysis using long-read sequencing on a PacBio Sequel system (PacBio, Menlo Park, CA, USA), were sequenced. These control samples harboured one of nine mtDNA variants of known heteroplasmy levels. The known mtDNA variant was visualised in BAM files in IGV to determine the levels of heteroplasmy.

## 3. Results

### 3.1. Summary of Genetic Findings

In total, 1352 samples were sequenced, and an overall average coverage of 85× was achieved for all smMIPs across all NovaSeq 6000 SP sequencing runs. Based on the smMIPs design, where, on average, each nucleotide is covered by eight smMIPs, we estimate a coverage of 680× per nucleotide. Seventy-four probands failed the sequencing analysis due to insufficient or low-quality DNA and remained genetically unexplained. Thus, a genetic explanation for disease was provided for 508 probands out of 1278 sequenced probands, obtaining a diagnostic yield of 39.8% (Appendix A). Among these probands, 459 different variants were found in 49 distinct genes and were deemed to be disease-causing, with 73 of these being novel variants not previously described in the literature, LOVD or ClinVar. These 459 variants included 249 missense, 63 frameshift, 60 nonsense, 56 splice-altering variants, 17 CNVs, 8 in-frame deletions, 3 start-lost variants, 1 in-frame insertion, 1 downstream gene variant and 1 5′ UTR variant (Appendix A; Figure 1). Overall, 379 probands were solved by variants in genes associated with autosomal recessive inheritance (74.6%); 105 probands were solved by variants in genes associated with autosomal dominance (20.7%); 18 probands were solved by variants in genes with an X-linked inheritance (3.5%); and 6 probands were solved by mitochondrial variation (1.2%). The top five most frequent causative genes in our cohort were *ABCA4* (37.2%), *PRPH2* (6.7%), *CDHR1* (6.1%), *PROM1* (4.3%) and *RP1L1* (3.1%), as depicted in Figure 2. Of the 508 probands, there were pathogenic variants in more than one gene for eight probands (Appendix A). In this instance, the genotype, which best correlated with the submitted phenotype upon consultation with the referring clinician, was selected as the primary cause of disease in each case.

Heterozygous missense VUSs were identified in six probands, which, by our criteria, should be designated as unsolved (Appendix A). Notwithstanding, we consider these variants to be likely causative in these probands, given previous evidence in the literature, which predicts them to be highly deleterious based on in silico pathogenicity predictions and low allele frequencies in genes associated with autosomal dominant MDs. Despite the lack of functional evidence to confirm the causality of these variants, the possibility that each variant might be disease-causing could not be excluded; thus, the six probands were considered “possibly solved”.

### 3.2. Variants with Putative Incomplete Penetrance

To explore whether the contribution of reportedly pathogenic variants to disease is as rare as expected and to predict penetrance, we compared the gnomAD-ALL AF of all variants identified in probands considered genetically solved with the AF within the iMD cohort and inferred penetrance. Of the 73 rare variants identified in the present study, all variants were considered rare, i.e., either absent from gnomAD or with a gnomAD AF ≤ 0.002%; therefore, we were unable to confidently estimate their penetrance. Ten variants, which were deemed to contribute to genetic diagnoses, were of particular interest, given the AFs and the estimated penetrance. Of these, eight have been reported in the literature as variants with reduced penetrance (*ABCA4* c.5603A>T, c.5882G>A and c.4253+43G>A; *NMNAT1* c.769G>A, *PRPH2* c.424C>T, c.514C>T and c.623G>A, *RP1L1* c.133C>T). We further highlight *CDHR1* c.783G>A and *CNGB3* c.1208G>A as variants with incomplete penetrance. These 10 variants were present at least once in 143 out of 508 probands considered solved (28.1%) (Appendix A). Consequently, probands harbouring these variants and considered genetically solved cannot be estimated with certainty.

Excluding *ABCA4* allele combinations, the most frequent variant in the solved cohort was the *CDHR1* c.783G>A variant, present in 31 out of 2704 alleles (AF 1.15%). Thirteen individuals were homozygous for the c.783G>A variant, and five individuals were presumed compound heterozygous, of which the second allele was c.143C>A in two individuals. Of the homozygous individuals, 12 out of 13 displayed a MD or STGD phenotype, and one proband presented with retinal pigment epithelium dystrophy (RPED). In the gnomAD genome and exome databases, *CDHR1* c.783G>A has an AF of 0.3052%, and c.143C>A has a frequency of 0.0383%. Thus, under a Hardy–Weinberg assumption, we expect the frequency of c.783G>A homozygotes to be 0.000931% and the frequency of c.783G>A/c.143C>A heterozygotes to be 0.000234%. We observed thirteen homozygotes in 1352 individuals and two c.783G>A/c.143C>A heterozygotes. Assuming an overall prevalence of iMDs as 1 in 5000, this suggests a penetrance of 20.7% for c.783G>A homozygotes and 12.6% for c.783G>A/c.143C>A heterozygotes. Since *CNGB3* c.1208G>A was observed even less frequently in our patient cohort than c.783G>A, and it had an even larger population frequency (0.4161%), its penetrance was lower still. Under a Hardy–Weinberg assumption, we expect the frequency of *CNGB3* c.1208G>A homozygotes to be 0.001731%. We observed four homozygotes in 1352 individuals; therefore, assuming an overall prevalence of iMDs as 1 in 5000, this suggests a penetrance of 3.4% for c.1208G>A homozygotes.

### 3.3. ABCA4-Associated Retinopathies

Of the 1352 probands, 1083 (80.1%) had undergone at least one type of genetic pre-screening prior to inclusion in this study (Appendix A). In total, 837 probands (61.9%) had previously undergone complete smMIPs sequencing of the *ABCA4* gene due to a STGD or STGD-like phenotype, where either one variant (*n* = 404) or no variants (*n* = 433) in *ABCA4* were identified [30,31]. Following sequencing with our MD-smMIPs panel, 233 (27.9%) of these samples are now considered genetically solved. Among these, bi-allelic *ABCA4* variants could genetically explain 10 STGD1 probands. Of the 269 samples, which had not undergone previous genetic screening, 157 (58.4%) are now considered solved. 

Three novel variants in *ABCA4*, which passed our filtering thresholds for SpliceAI, were selected for in vitro splice assays (Appendix A). The c.3329-124G>T variant was identified in proband 071829 (SpliceAI delta scores: AG 0.43, AL 0.21) and was assessed using a midigene splice assay. Gel analysis and Sanger sequencing revealed a band of 831 nt corresponding to the WT construct and 940 nt corresponding to 109 nt of intron 22 retention, leading to a frameshift (r.[3328_3329ins3329-109_3329-1,=]; p.[Gly1110Valfs*4,=]) (Figure 3A). For the mutant construct, the WT band was present at 45%; however, 55% of the RNA showed intron retention of 109 nt. Based on the percentage of aberrant splicing observed [31], c.3329-124G>T is considered a mild variant (>40% and <80% WT RNA). The c.1451A>G variant in proband 067130 (SpliceAI scores: AG 0.94, AL 0.21) is a missense variant, classified as a VUS; yet, the splicing prediction tools predicted a loss of 95 of the 198 nt of *ABCA4* exon 11. This variant was also assessed in a midigene system, which confirmed the SpliceAI predictions of a novel splice acceptor site (SAS) within exon 11, leading to partial (95 nt) exon skipping (Figure 3B). Given the percentage of WT RNA remaining for this variant (62%), it is also considered a mild variant (r.[=,1357_1451del]; p.[Lys484Arg,Asp453Glyfs*8]). In this proband, this variant was present with a second severe *ABCA4* allele; therefore, if confirmed to be in *trans*, this will provide a genetic explanation for the disease for this patient. Variant c.6817-679C>G was identified in proband 066668, alongside a second *ABCA4* DIV c.4539+2028C>T, previously determined to have a moderately severe effect [32,33,34]. We already considered a novel homozygous *RP1L1* variant resulting in a frameshift (c.1509del; p.(Gly504Alafs*4)) to be the genetic explanation for disease for this patient, but we could not fully exclude the involvement of *ABCA4* without further exploration of this novel DIV due to a previous report of an alternate allele, which was splice-altering (c.6817-679C>A, p.Gln2272_Asp2273fs*10, r.[=,6816_6817ins6816+1_6817-682]) (SpliceAI delta scores: AG 0.70, DG 0.91) [35]. Despite passing our SpliceAI filtering thresholds, c.6817-679C>G showed no splicing effect when tested in a novel minigene construct in HEK293T cells (Figure 3C). Moreover, together with the aforementioned *ABCA4* splice-altering candidate variants, nine of the novel variants identified are predicted to impact splicing (Appendix A). 

### 3.4. CACNA1F Variant Female Carrier 

Among the 73 novel variants, we identified the heterozygous *CACNA1F* c.1566_1575del; p.(Trp533Cysfs*9) variant in a female patient with MD (proband 079822). *CACNA1F* is associated with X-linked CRD [36], and c.1566_1575del results in a frameshift, which is predicted to lead to a protein truncation. The female proband in the current study was examined at 47 years of age and described visual disturbance since the age of 45. Macular, small hypopigmented dots were present in both eyes; visual acuity (VA) was 6/6 BE; visual field (VF) was normal; and a full-field electroretinogram (ffERG) demonstrated normal rod response and mildly attenuated cone response. Multifocal ERG (mfERG) showed that the responses were marginally reduced relative to normal. The clinical diagnosis for this patient was inconclusive, which may be in line with a mild phenotype associated with an X-linked carrier phenotype [37,38]. Thus, we consider this patient to be possibly solved.

### 3.5. New Genotype–Phenotype Correlations

Many probands presenting with STGD could be solved by variants in other iMD-associated genes. In some instances, variants were identified in genes that are typically associated with a different phenotype. Where possible, additional clinical information was obtained to re-evaluate and re-classify genetic findings. In one proband (proband 079830) with a central areolar choroidal dystrophy (CACD) diagnosis, we identified compound heterozygous variants in *ROM1*—c.339dupG; p.(Leu114Alafs*18) and c.712del; p.(Leu238Cysfs*78)—both of which result in a frameshift. Based on their co-occurrence pattern in gnomAD (https://gnomad.broadinstitute.org/variant-cooccurrence?dataset=gnomad_r2_1 (accessed on 1 March 2023)), these variants are likely found on different haplotypes in most individuals, rendering this combination of variants ultra-rare. For this patient, haplotype phasing using PacBio long-read amplicon-based sequencing confirmed that the variants are in *trans*, providing evidence that this novel combination of rare variants is likely causative (Appendix A). 

### 3.6. Carriers of (Likely) Pathogenic Variants

Of the 508 probands considered solved, 134 (26.4%) had at least one additional ACMG class 4 or 5 mono-allelic variant, it being found 1) in a gene associated with autosomal recessive inheritance or 2) a variant in *ABCA4* classified as mild, moderately severe or severe (Appendix A). In the genetically unexplained cohort, a class 4 or 5 mono-allelic variant, or a variant in *ABCA4* classified as mild, moderately severe or severe, was identified in 322 (38.2%) probands (excluding *ABCA4* c.2588G>C, which is benign when not in *cis* with c.5603A>T [13]) (Appendix A). 

### 3.7. Mitochondrial DNA Variants

In the 13 mtDNA positive controls, the levels of heteroplasmy for known mtDNA variants associated with MDs and other mitochondrial disorders could be called with 99% concordance (Lin’s coefficient of concordance ρc= 0.9962; 95% confidence interval from 0.9879 to 0.9988). Five of the thirteen positive controls harbouring known mtDNA heteroplasmic SNPs carried the m.3243A>G variant implicated in macular pattern dystrophy [29], which was called accurately in four controls by variant callers UnifiedGenotyper and HaplotypeCaller from GATK (16–42% heteroplasmy). One positive control had 6% heteroplasmy, which could be determined upon manual assessment of sequencing read alignments, but it was not called by the variant callers. In the iMD cohort, m.3243A>G was called in six iMD probands, with heteroplasmy levels ranging from 13 to 31%.

## 4. Discussion

Here, we further utilised the MD-smMIPs panel previously described [22] to sequence 1352 additional probands diagnosed with iMDs to provide a genetic diagnosis for 508 patients. Nevertheless, we consider that, given the reduced penetrance of at least nine variants, one-third of these 508 patients (28.0%; *n* = 142) may in fact remain partially unsolved.

### 4.1. Reduced Penetrance of Variants

Of the probands we consider genetically solved, there may be other implicating factors, which—when scrutinised—may reduce our overall diagnostic yield due to potential incomplete penetrance. Using an estimated prevalence of iMDs (1 in 5000 individuals) and gnomAD-ALL AFs, 10 variants are proposed as examples of reduced penetrance. In addition to previously reported variants in *ABCA4* (*n* = 3), *NMNAT1* (*n* = 1), *PRPH2* (*n* = 3) and *RP1L1* (*n* = 1) [1,2,3,5,6,8,9,10,11,12], we highlighted putative incomplete penetrance for two additional variants: *CDHR1* c.783G>A and *CNGB3* c.1208G>A. These variants are present in the general population at a higher frequency than expected, casting doubt regarding the pathogenicity of these variants alone as the cause of disease for all cases. For example, *CDHR1* c.783G>A has a high AF in the general population (AF: 0.3052%, which was observed at the highest frequency of 0.5891% in the European-Finnish population) (gnomAD, 17 April 2023). The c.783G>A; p.Asp214_Pro261del splice-site variant, located at the last nucleotide of exon 8, is considered a hypomorphic variant, meaning that some functional *CDHR1* protein is still produced. Reduced protein levels may cause IRD in patients due to the critical role of *CDHR1* in the structure and function of photoreceptors [39]. Natural skipping of exon 8 of *CDHR1* has been described, which is strengthened by c.783G>A [40]. Variants in *CDHR1* result in variable phenotypes [40,41,42]. However, this variant is considered pathogenic and results in a mild phenotype [40]. Similarly, the *CNGB3* c.1208G>A; p.(Arg403Gln) variant has been reported in patients with highly variable macular phenotypes in conjunction with *CNGA3* digenic inheritance [43]. Variants in the *CNGB3* gene are responsible for approximately 50% of all patients with achromatopsia [44] and not iMD, which complicates this analysis. Coupled with a gnomAD AF of 0.4164%, which was observed at the highest frequency in the south Asian population, at 2.7315% (Genome Aggregation Database, 17 April 2023), it cannot be excluded that this variant displays incomplete penetrance. 

A large proportion of the present cohort (*n* = 837; 61.9%) had previously undergone complete smMIPs sequencing of the *ABCA4* gene due to a STGD or STGD-like phenotype [30,31]. Consequently, we considered 27.9% (*n* = 233) of these probands genetically solved; however, based on a more conservative assessment, allele combinations including mild *ABCA4* variants with reduced penetrance cast doubt as to whether these combinations fully explain disease causality in all probands. For example, the c.769-784C>T variant is a major *cis*-acting modifier of the c.5882G>A allele, contributing to its penetrance [45], but it is not assumed to be a causal (mild) variant when present alone in one allele. The possibility that non-*ABCA4* genetic—or non-genetic—factors may influence these mild alleles cannot be excluded.

We hypothesise that approximately 70% of the “solved” cases in this study are explained with fully penetrant variants or, if in recessive genes, fully penetrant combinations of variants. It is important to note that our penetrance calculations are based on an assumption of the prevalence of iMDs, not taking into account the frequency of the variants in distinct population groups or considering unidentified potential genetic modifiers. A higher or lower disease prevalence than our estimate would impact actual penetrance. Importantly, we also did not perform a segregation analysis for all probands; thus, we assume that the identified variants are in *trans*. Prevalence estimates have been reported for CRDs, which are estimated to affect 1 in 30,000–40,000 individuals worldwide [46], and STGD1, which is estimated to affect 1 in 19,000–22,000 individuals in the Netherlands [47]. Given that there are no data reporting an accurate prevalence of iMDs as a whole, we consider the 1 in 5000 total prevalence of iMDs as our best estimate. Thus, our calculations do not report the exact penetrance rates, but they highlight putative incompletely penetrant variants, which should be interpreted with caution when making a genetic diagnosis.

### 4.2. Putative Digenic Inheritance

Although combinations of variants are known for some genes, there are still limited data to draw conclusions regarding digenic inheritance in IRDs. In this cohort, we do not have sufficient sample numbers to identify strong connections between genes and potential digenic/complex inheritance, except for those already reported. Thus, such few cases cannot be analysed statistically, and these cases might represent co-occurrence of variants simply by chance. Knowledge of digenic inheritance is recognised for *CNGA3* and *CNGB3*, where we observed the previously reported homozygous *CNGB3* p.(Arg403Gln) variant with a heterozygous pathogenic *CNGA3* allele, p.(Val529Met) [43]. In addition, digenic or tri-allelic inheritance has been reported, with patients carrying *PRPH2* p.(Arg172Trp) with a *ROM1* and/or *ABCA4* variant. The phenotypic severity of patients carrying the *PRPH2* variant increased with the addition of a variant in *ROM1* [48]. We observed co-occurrence of an alternative variant in *PRPH2*, p.(Arg172Trp), with a *ROM1* variant, p.(Leu114Alafs*18), as well as *PRPH2* p.(Arg142Trp) present with *ABCA4* p.(Arg24His), which is categorised as a mild variant. Since p.(Arg142Trp) shows reduced penetrance, it is plausible that the addition of even a mild *ABCA4* allele may still contribute to a variable phenotype or age of onset. Furthermore, we highlighted eight probands who have potential disease-associated variants in more than one gene (Appendix A). There are insufficient data in this study to draw conclusions on whether, in these instances, (1) digenic inheritance is displayed; (2) the combination of variants contributes to the phenotype; or (3) the confirmation of the causative variants can aid in reclassification of other variants, which are present.

### 4.3. New Genotype–Phenotype Associations

Knowledge of phenotypic and genetic heterogeneity in MDs has been previously described [7,8,49,50,51], which can complicate a clear diagnosis and patient outcomes. In the present study, many probands presenting with STGD could be solved by variants in other iMD-associated genes, further emphasising the genetic heterogeneity of MDs and the existence of STGD phenocopies. Compound heterozygous variants in *ROM1* (c.339dupG; p.(Leu114Alafs*18) and c.712del; p.(Leu238Cysfs*78)) were identified in a patient with CACD (proband 079830). Variants in *ROM1* have previously been associated with digenic RP, with p.(Leu114Alafs*18) reported to manifest with *PRPH2* variants, namely p.(Leu185Pro) [52,53] and p.(Arg142Trp) [54,55]. In fact, the presence of p.(Leu114Alafs*18) with p.(Arg142Trp) in a patient with autosomal dominant CACD has previously been described [54]. Moreover, rare variants in *ROM1* and a common *PRPH2* haplotype are proposed to be the modifiers of *ABCA4*-associated disease [56]. Despite previous examples of digenic inheritance of *ROM1* and *PRPH2*, no *PRPH2* or *ABCA4* variants were identified in proband 079830, although high sequencing coverage was observed over the targeted regions. Furthermore, a segregation analysis could not be performed for this proband to phase the variants; therefore, long-read amplicon sequencing was performed. Long-read amplicon sequencing confirmed that these variants are in *trans*; thus, to our knowledge, this is the first incidence in the literature of this combination of *ROM1* bi-allelic variants, although knowledge of p.(Leu238Cysfs*78) in a homozygous manner has been described in late-onset pattern maculopathy [57]. This finding warrants further exploration to understand the contribution of *ROM1* to disease pathology in CACD probands.

Despite the genetic variability observed, phenotypic overlap exists between AMD and iMD (AMD-mimicking dystrophies), which can lead to misdiagnoses [9,58,59]. Genetic screening for MD-associated genes in patients with AMD discovered seemingly Mendelian variants, including a *PRPH2* variant, as well as enrichment of heterozygous *ABCA4* variants [9]. In addition, monogenic inheritance of variants in the complement factor H (*CFH*) gene has been reported, whereby individuals carrying the *CFH* p.(Tyr402His) AMD risk allele together with a *CFH* null allele have an increased risk of developing early-onset basal laminar drusen [60]. Furthermore, rare and common variants in *ABCA4*, *FBN2* and *TIMP3* have been shown to contribute to iMD and AMD [61,62,63,64]. Null variants in genes associated with AMD may be associated with an iMD phenotype, which in turn may also be challenging in clinical distinction, for example, between late-onset iMDs or early-onset AMD, since patients with iMD subtypes can show characteristics associated with AMD [9,58,59]. By focusing our analysis on rare variants in iMD-associated genes and rare variants with a high predicted pathogenicity in AMD genes, we consider one proband possibly solved by a novel heterozygous variant in *HMCN1*: c.16304A>C; p.(Glu5435Ala). Previous evidence supports the implication of *HMCN1* in early-onset AMD, whereby *HMCN1* variants may demonstrate incomplete penetrance or confer susceptibility to disease [65,66,67]. Further studies are required to better understand the role of *HMCN1* in iMDs.

### 4.4. Mitochondrial DNA Variants

Using the MD-smMIPs panel, mitochondrial DNA could be reliably sequenced, and heteroplasmic SNPs could be determined with high concordance in positive controls, including the retinal-associated m.3243A>G variant. The mitochondrial variant m.3243A>G is associated with a wide spectrum of clinical conditions, such as mitochondrial myopathy, encephalopathy, lactic acidosis, stroke-like episodes (MELAS) syndrome, diabetes and deafness, as well as retinal pigment abnormalities encompassing macular pattern dystrophy and mitochondrial retinal dystrophy [29,68,69,70,71]. It is difficult to determine a “critical” threshold of heteroplasmy above which m.3243A>G is pathogenic, since the degree of heteroplasmy in different tissue types does not appear to be correlated with the severity of macular dystrophy [29]; therefore, follow-up studies in relevant tissues would be required. Despite an expected lower heteroplasmy in blood than in retina, next-generation sequencing technologies can be used for accurate detection of heteroplasmy in blood, including low-level heteroplasmy, if sufficient read coverage is obtained. However, a negative result in blood does not exclude the presence of a mutant mtDNA in the tissue expressing the disease, in which the heteroplasmy level is higher. Individuals with the m.3243A>G variant at known levels of heteroplasmy were included as positive controls; however, those with <16% heteroplasmy were not called by variant callers, and thus, additional cases may have been missed. Since no mitochondrial purification steps were performed prior to the library preparations, the homology between some mtDNA and nuclear DNA remains a complicating factor, which may result in difficulty in alignment and variant calling. However, smMIPs were balanced to reduce the overall mtDNA reads and provide a more even representation of the mtDNA and nuclear DNA targets to maximise read coverage across all targets [22]. We suggest that individuals with the m.3243A>G variant should be re-assessed by an ophthalmologist to determine whether the retinal abnormalities are consistent with this genetic finding. Furthermore, there is clinical overlap between iMDs, AMD and mitochondrial macular abnormalities [29]; thus, mitochondrial variation should be considered in MD screening panels.

### 4.5. Limitations

The terms “very likely solved” and “possibly solved” were used to provide a final verdict for each proband following smMIPs-based sequencing analysis. However, these should be interpreted with caution, particularly the “possibly solved” verdict based on identification of two VUSs. Segregation analysis was not routinely performed; thus, compound heterozygous variants are presumed to be in *trans*. Penetrance calculations were based on an estimation of iMD prevalence in the general population using the estimated overall reported prevalence for CD, CRD and STGD1 [46,47,72]; thus, our penetrance estimates should be taken as penetrance for “all forms of iMD in the general population”. If any variant is specific for a specific subtype, the prevalence of that subtype is overestimated, and the penetrance of the variant is underestimated. On the other hand, we used the global estimates of allele frequencies, and if allele frequency varied considerably among people of different ancestry, our Hardy–Weinberg assumption may have underestimated the true frequency of homozygotes, and therefore, we may have overestimated the penetrance of these homozygous and compound heterozygous genotypes. Furthermore, in instances of compound heterozygosity, we made no inference concerning the *cis*–*trans* relationship of the two alleles, and this is certainly likely to be important. Overall, while we are highly confident in the inference of incomplete penetrance, the precise level of that penetrance is only our best estimate. Another caveat is that the phenotype may not be concordant with the genotype findings upon re-assessment. Therefore, probands with this status must be assessed carefully to ensure that (1) the phenotype matches the genetic findings and (2) the variants are confirmed in *trans* following segregation analysis. Similarly, probands with combinations of variants in *ABCA4* must also be assessed with care. Different *ABCA4* variants result in different levels of remaining protein activity; therefore, different combinations of variants result in several retinal phenotypes, clinically manifesting from mild to severe [73]. When an *ABCA4* allele is identified, the severity categories based on residual ABCA4 protein activity are taken into account [11,73,74]. When variants are (presumed) in *cis*, the most severe of the two variant severities is used (e.g., an allele with mild and moderately severe variants in *cis* would be given a severity category of moderately severe). When the mild *ABCA4* variant c.5882G>A; p.(Gly1961Glu) is present in *trans* with a second allele, which is considered mild–moderately severe, moderately severe or severe in a patient with a STGD1 phenotype, and no other putative iMD gene variants are identified, we also used the term “possibly solved”. However, we cannot exclude the possibility that other variants in the intronic regions of *ABCA4*, which were not sequenced in this panel, or genetic modifiers in other genes, may be at play.

Bi-allelic *ABCA4* variants were identified in 10 patients, which were mono-allelic, following previous smMIPs sequencing of the *ABCA4* gene [23]. The design of the smMIPs in this study captures a larger target region of 225 nt [22], more than two times that of the previous smMIPs design [23]. In addition, there is an increase in the number of independent smMIPs in the MD-smMIPs panel design, with an average of eight smMIPs covering each nucleotide versus up to two smMIPs in the earlier design [23]. Taken together, this increase enables variants to be detected with higher confidence due to multiple smMIPs covering genomic regions (i.e., higher coverage). For this reason, we could detect variants in *ABCA4*, which were previously overlooked using double-tiled smMIPs. Moreover, the advent of new deep-learning tools for predicting the impact of potential splice-altering variants, including SpliceAI [24], has aided in the identification of novel putative splicing variants. SpliceAI has shown to be one of the best performing tools for predicting the impact of DIVs in *ABCA4* [75]; thus, the utilisation of this tool contributed to novel *ABCA4* variants with high SpliceAI delta scores being detected, which were not previously identified and confirmed using splice assays.

Despite our efforts, at least 844 probands (60.2%) remained genetically unsolved following smMIPs-based sequencing of MD-associated genes and loci, with one likely pathogenic or pathogenic variant identified for 38.2% (*n* = 322) of the unsolved cohort. The mono-allelic variant may contribute to disease in these cases, whereby a second variant may not have been captured or missed in our analysis. CNV detection can be cumbersome if the average sequencing coverage is low, and inversions cannot be detected; thus, unidentified CNVs could contribute to causality for this subset of patients. Moreover, novel DIVs outside of the targeted regions cannot be detected. However, this percentage is in line with a previous study, highlighting that 36% of the general population are healthy carriers of at least one recessive IRD allele [76]; thus, we would expect the same in all IRD cases. Finally, utilising a targeted smMIPs-sequencing approach holds the caveat of only covering MD-associated genes; therefore, WES or WGS is still warranted for the genetically unexplained subset.

## 5. Conclusions

Although 39.8% of iMD probands were deemed genetically solved following smMIPs sequencing, almost one-third of these patients could not be solved solely by reported variants with complete penetrance or fully penetrant combinations of variants in genes associated with a recessive IRD. Here, we highlight at least 10 variants with suggested reduced penetrance in iMDs. Additional intronic *cis*-modifiers or modifiers in other genes or non-genetic modifiers may remain to be identified; therefore, additional sequencing methods, such as WGS, are still warranted for genetically unexplained probands. The identification of novel variants suggests that many rare variants remain undiscovered in iMD cohorts, where the use of smMIPs technologies may solve some of the missing heritability for iMDs in a cost-effective manner. Furthermore, revealing novel splice-altering variants offers the opportunity for splice-modulation therapies to be employed for patients. The identification of rare variants in genes associated with AMD highlights the challenges that arise when making a diagnosis. Overall, this emphasises the need for further research into the underlying molecular mechanisms, which lead to macular dystrophies.

## Figures and Tables

**Figure 1 biomolecules-14-00367-f001:**
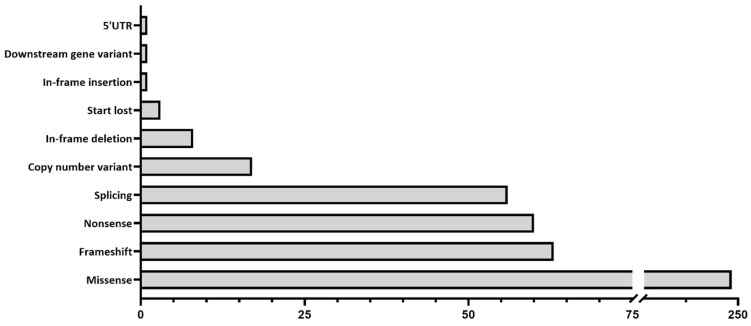
Types of variants identified in the 508 individuals, where a genetic diagnosis was considered identified.

**Figure 2 biomolecules-14-00367-f002:**
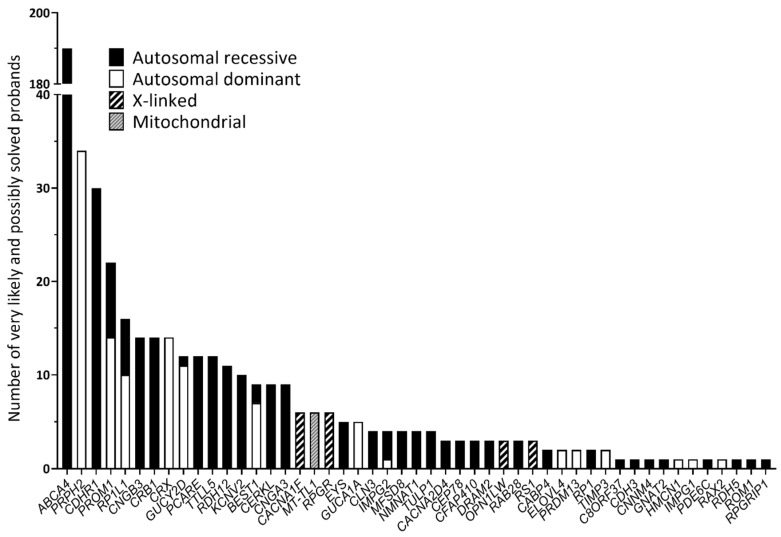
Genes involved in the 508 individuals considered very likely or possibly solved following genetic analysis.

**Figure 3 biomolecules-14-00367-f003:**
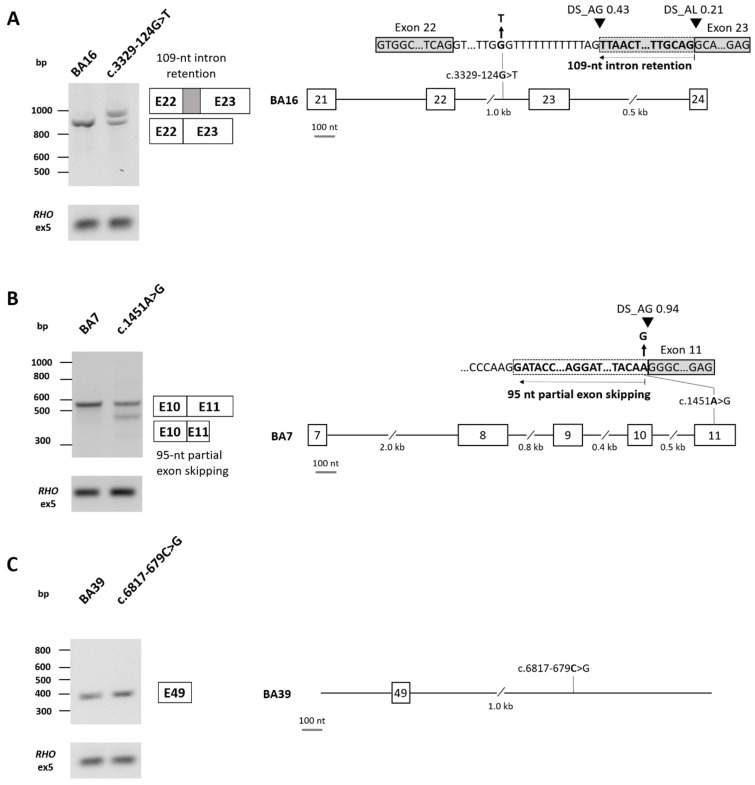
*ABCA4* splicing assay results in HEK293T cells. (**A**) Variant c.3329-124G>T in intron 22 was tested in a midigene system using the BA16 construct, which incorporated exons 21–24 of *ABCA4*. Agarose gel analysis and Sanger sequencing revealed a band of 831 nt corresponding to the wild-type (WT) construct and 940 nt corresponding to 109 nt of intron 22 retention, leading to a frameshift (r.[3328_3329ins3329-109_3329-1,=]; p.[Gly1110Valfs*4,=]). Based on the percentage of aberrant splicing observed (45% of WT RNA remaining), c.3329-124G>T is considered a mild variant. Original images can be found in Appendix A. (**B**) Variant c.1451A>G in exon 11 was tested in a midigene system using the BA7 construct, which incorporated exons 7–11 of *ABCA4*. Agarose gel and Sanger sequencing analysis revealed a band of 542 nt corresponding to the WT construct, in addition to a band of 448 nt confirming a loss of 95 of the 198 nt of exon 11, resulting in partial exon skipping. A faint band was also observed at 331 nt, corresponding to exon 10 skipping with partial exon 11 skipping; however, this band contributed to <15% of the total RNA and was thus not considered in the calculations. Based on the percentage of aberrant splicing observed (62% of WT RNA remaining), c.1451A>G is considered a mild variant (r.[=,1357_1451del]; p.[Lys484Arg,Asp453Glyfs*8]) Original images can be found in Appendix A. (**C**) Variant c.6817-679C>G in intron 49 was tested in a minigene system using the BA39 construct. Agarose gel and Sanger sequencing showed only the WT construct (361 nt; r.(=); p.(=)). Original images can be found in Appendix A.

## Data Availability

The data that support the findings of this study are available from the corresponding author upon reasonable request.

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
