# Peer review of "Towards Uncovering the Role of Incomplete Penetrance in Maculopathies through Sequencing of 105 Disease-Associated Genes"

_biomolecules, 2024, doi:10.3390/biom14030367_

Round 1
Reviewer 1 Report (Previous Reviewer 1)
Comments and Suggestions for Authors
revised draft is improved and the concerns responded satisfactorily.
Author Response
The authors would like to thank the reviewer for their time taken to read the revised manuscript. We are very grateful for your feedback.
Reviewer 2 Report (New Reviewer)
Comments and Suggestions for Authors
The paper lacks novelty, this is an extension of the work by the group published in the Hum Mutat 2022. As author mentioned “In total, 837 probands 302 (61.9%) had previously undergone complete smMIPs sequencing of the ABCA4 gene due to a STGD or STGD-like phenotype, where either one or no variants in ABCA4 were identified [21,22]. Following sequencing with our MD-smMIPs panel, 233 (27.9%) of these samples are now considered genetically solved. Amongst these, bi-allelic ABCA4 variants 306 could genetically explain 10 STGD1 probands.” I'm uncertain why the ABCA4 mutation was not detected in smMIPs sequencing in the previous study, as revealed by this study.
Towards uncovering the role of incomplete penetrance in maculopathies through sequencing of 105 disease-associated genes.
1. Line 35, 37, 42 – insert references
2. Line 97 – Elaborate on selection criteria. What were the clinical parameters/ imaging techniques used to make a diagnosis for iMD? Where these uniform across all 31 institutes? What measures were taken to ensure uniformity in recruitment?
3. Line 250 – Expand in discussions or results section
4. Line 304 – mention segregated proportions of patients that were previously identified
5. Line 434 – Your classification system of ‘solved, unsolved’ has not been elaborated in the text. Please mention inclusion/ exclusion criteria for categorisation.
Comments on the Quality of English Language
The overall flow of information in the manuscript could be improved for better comprehension by a broad audience, ranging from clinicians to scientists. The authors should revise the manuscript to simplify language and present the content in a more accessible manner.
Author Response
The authors would like to thank the reviewer for their time to read our manuscript and provide comments. We have tried our best to address the reviewer’s comments, as below:
- Line 35, 37, 42 – insert references
Author response: additional references have been included in the introduction.
- Line 97 – Elaborate on selection criteria. What were the clinical parameters/ imaging techniques used to make a diagnosis for iMD? Where these uniform across all 31 institutes? What measures were taken to ensure uniformity in recruitment?
Author response: this has been elaborated, as follows:
iMD and cone-dominated patients were selected if a clinical diagnosis of STGD, a “STGD-like” phenotype, MD or a cone-led retinal degeneration (e.g. CD or CRD) was made by the referring senior ophthalmologist at each institute, based on a patient medical history, their family history and a detailed ophthalmological exam. According to the clinical practice and available equipment of each institute, where possible, most of the following tests were evaluated and performed: best-corrected visual acuity, colour vision test, visual field test, slit lamp fundus examination, colour fundus imaging, fundus autofluorescence (FAF) imaging, optical coherence tomography (OCT) imaging, microperimetry and/or electroretinography (ERG).
- Line 250 – Expand in discussions or results section
Author response: the following has been added to this sentence in the results section:
In this instance, the genotype that best correlated with the submitted phenotype, upon consultation with the referring clinician, was selected as the primary cause for disease in each case.
We also expanded on this in the ‘Putative digenic inheritance’ section, as follows:
Furthermore, we highlighted eight probands that have potential disease-associated variants in more than one gene (Supplementary Table 4). There are insufficient data in this study to draw conclusions on whether, in these instances, 1) digenic inheritance is displayed, 2) the combination of variants contribute to the phenotype, or 3) the confirmation of the causative variants can aid in the reclassification of other variants that are present.
- Line 304 – mention segregated proportions of patients that were previously identified
Author response: this has been included, as follows:
In total, 837 probands (61.9%) had previously undergone complete smMIPs sequencing of the ABCA4 gene due to a STGD or STGD-like phenotype, where either one variant (n = 404) or no variants (n = 433) in ABCA4 were identified [27,28].
- Line 434 – Your classification system of ‘solved, unsolved’ has not been elaborated in the text. Please mention inclusion/ exclusion criteria for categorisation.
Author response: a small addition has been made to the following sentence (was lines 165-168), since this categorisation was described previously in the literature:
A final verdict of “very likely solved’’, “possibly solved’’ or “unsolved’’ was assigned, as previously described[19], considering the allele frequencies, suggested ACMG classification, pathogenicity of the variants, the gene(s) involved, and previous reports in online databases.

Reviewer 3 Report (New Reviewer)
Comments and Suggestions for Authors
The current paper is an exciting contribution to the field of inherited maculopathies. The sample size is big, the methodology is sound and the results are interesting.
The reviewer has minor points:
Many sentences especially in the introduction lack references.. sometimes one complete paragraph could lack the references (first paragraph).
Gene symbols are not always in italics throughout the text.
The readers of this paper could benefit from discussing previous work showing similar results. For instance, the paper of Audo et al published in IJMS three years ago analyzed two main maculopathies (CD and/or CRD) and showing that ABCA4, PRPH2 are major genes having specific phenotypic signatures (https://www.mdpi.com/1422-0067/20/19/4854).
Author Response
The authors would like to thank the reviewer for taking the time to consider our manuscript and for the useful feedback. We have tried our best to address the reviewer’s comments, as below:
Many sentences especially in the introduction lack references.. sometimes one complete paragraph could lack the references (first paragraph).
Author response: additional references have been included in the introduction.
Gene symbols are not always in italics throughout the text.
Author response: we thank the reviewer for highlighting this and have made the necessary amendments.
The readers of this paper could benefit from discussing previous work showing similar results. For instance, the paper of Audo et al published in IJMS three years ago analyzed two main maculopathies (CD and/or CRD) and showing that ABCA4, PRPH2 are major genes having specific phenotypic signatures (https://www.mdpi.com/1422-0067/20/19/4854).
Author response: we would like to thank the reviewer for highlighting this and the above publication. The authors have added the following the ‘New genotype-phenotype associations’ section:
Knowledge of phenotypic and genetic heterogeneity in MDs has been previously described[45-47], which can complicate a clear diagnosis and patient outcomes. In the present study, many probands presenting with STGD could be solved by variants in other iMD-associated genes, further emphasizing the genetic heterogeneity of MDs and the existence of STGD-phenocopies.
This manuscript is a resubmission of an earlier submission. The following is a list of the peer review reports and author responses from that submission.
Round 1
Reviewer 1 Report
Comments and Suggestions for Authors
The authors have screened a significant number of patients to identify potential genetic variants in 150 genes and claimed to solve around 40% of cases. The introduction is comprehensive, and the methodology is current; however, there are a few concerns.
An estimation of the prevalence of 172 iMDs in the population (1 in 5,000) needs reference.
What about the penetrance of rare novel variants identified in this study?
The ethnicity of patients has not been considered when penetrance is estimated. The ethnic biases of inherited variants or the presence of modifiers need to be well explained before solving genetic causes of disease in patients.
The estimation of penetrance of certain alleles in a specific population, based on its general AF, needs strong evidence; the estimation of controls matching ethnicity may be helpful. Moreover, the contribution of an allele having reduced penetrance may not be overestimated in the case of compound heterozygotes.
Author Response
The authors would like to thank the reviewer for taking the time to read our manuscript and provide constructive feedback. We have tried our best to address the reviewer’s comments, as below:
Reviewer comment: An estimation of the prevalence of iMDs in the population (1 in 5,000) needs reference.
Author response: The authors attempted to find a study reporting the estimated prevalence of all iMDs; however, there are no reports in the literature estimating the prevalence of all these conditions together. We tried to address this in the discussion (original submission, lines 454-460) but have retrospectively calculated an estimate of the prevalence of iMDs based on the following prevalence figures reported in the literature; however, this is not an exhaustive list and excludes some inherited macular dystrophies, e.g. Sorsby fundus dystrophy, North Carolina macular dystrophy, for which worldwide prevalence are not reported:
- Best vitelliform macular dystrophy = 1 in 16,500 - 1 in 21,000 in at Olmsted County, Minnesota (Dalvin et al., 2016; PMID: 27120116); other online resources state 1 in 10,000 worldwide, yet with no citations.
- Cone dystrophy = 1 in 20,000 – 1 in 40,000 worldwide (Kohl, 2009; PMID: 19184602)
- Cone-rod dystrophy = 1 in 40,000 individuals worldwide (Hamel, 2007; PMID: 17270046)
- Stargardt disease = 1 in 8,000-10,000 (Tanna et al., 2017; PMID: 27491360)
Calculating the overall prevalence:
Overall prevalence = 1/ ((1/15000) + (1/30000) + (1/40000) + (1/9000))
Overall prevalence ≈ 1 in 4,235 ≈ 1 in 5,000 was our best estimate considering all iMDs.
We have included the following in the ‘Limitations’ section of the manuscript (lines 550-554) to show this: “Penetrance calculations were based on an estimation of iMD prevalence in the general population using the estimated overall reported prevalence for CD, CRD and STGD1 [36,37,60]. Thus, our estimation of 1 in 5,000 may represent an overrepresentation of the actual penetrance of variants, further supporting the hypothesis that these variants are not fully penetrant.”
Reviewer comment: What about the penetrance of rare novel variants identified in this study?
Author response: We calculated the penetrance of all rare novel variants identified in the study in the same way as described for those variants highlighted as variants of incomplete penetrance (using gnomAD-ALL allele frequencies (AFs) and the estimated prevalence of iMDs of 1 in 5,000). Of the 73 novel variants, 61 have an gnomAD-AF of 0 and therefore are not reported in the gnomAD populations, thus we expect these to be rare, fully penetrant variants (when inherited in the correct manner for a dominant or recessive condition). Of the 12 rare variants with an gnomAD-AF >0, AFs range from 0.000398% - 0.012909% and were present in the current study at frequencies n = 1 and therefore not highlighted as reduced penetrance in the study. An example is the PCARE c.2334T>G; p.(Tyr778*) variant, which has a gnomAD ALL AF of 0.0004%. PCARE c.2334T>G is estimated to be homozygous in 1 in 62,500,000,000 individuals in the gnomAD control population, thus if fully penetrant in a homozygous manner, we would expect to find 0.0108 individuals homozygous in the iMD study. We observed one homozygote and one compound heterozygote, both of which have an iMD phenotype. Using the penetrance calculation, this variant has an estimated penetrance 924,600%, which is an overinflated percentage because population-based estimates of the penetrance based on one patient are inaccurate. We have included an additional sentence in the results section (lines 272-275) as follows: “Of the 73 rare variants identified in the present study, 61 variants were absent from gnomAD exome and genome databases, with an AF of 0.00. Of the 12 variants with a gnomAD-AF greater than 0.00, AFs range from 0.000398% - 0.012909% and we considered these variants to be fully penetrant.”
Reviewer comment: The ethnicity of patients has not been considered when penetrance is estimated. The ethnic biases of inherited variants or the presence of modifiers need to be well explained before solving genetic causes of disease in patients.
Author response: Thank you for highlighting this concern, we agree that this has not been considered. The patients included in this study were collected from multiple international centres, for which ethnicity was not reported upon sample submission. To retrospectively collect these data would be outside of the scope of this study. However, we have included a sentence to highlight this in the limitations section (lines 554-558) as follows: “The ethnicities of patients were not considered in these penetrance calculations, which poses as another limitation to the study. Thus, estimated penetrance of variants may be affected by ascertainment bias. Furthermore, in instances of compound heterozygosity, when a variant with incomplete penetrance is in trans with a severe or null allele, its penetrance may be high”
Reviewer comment: The estimation of penetrance of certain alleles in a specific population, based on its general AF, needs strong evidence; the estimation of controls matching ethnicity may be helpful. Moreover, the contribution of an allele having reduced penetrance may not be overestimated in the case of compound heterozygotes.
Author response: A sentence has been added to the limitations section (lines 556-558): “Furthermore, in instances of compound heterozygosity, when a variant with incomplete penetrance is in trans with a severe or null allele, its penetrance may be high.”
Reviewer 2 Report
Comments and Suggestions for Authors
Just a few typos for correction:
line 153 - "the" is duplicated
line 240 - delete "been"
line 436 - Change to read "A large proportion of the present cohort"
line 461 - data is plural so should read "there ae still limited data"
Author Response
The authors would like to thank the reviewer for their time to read our manuscript and provide comments, in particular for making us aware of these errors in the text. We have implemented these changes in the revised manuscript.
We agree that the lack of segregation analysis for patients considered solved is a limitation to the study, so we have emphasized this by including a sentence in the abstract as follows: “Notably, segregation analysis was not routinely performed for variant phasing, a limitation that may also impact the overall diagnostic yield”. We already included the following sentence in lines 161-165 in the “Variant selection” section of the manuscript: “Segregation analysis was only performed for a small proportion of probands by individual collaborators, though it is an essential criterion according to the full ACMG guidelines; when at least two different rare variants in a single gene were detected in an individual, they were considered present in a bi-allelic state, and thus presumed compound heterozygous.”
As suggested, we have also amended the section “Considerations” to “Limitations”. Thank you again for your time.